# TorSeq: Torsion Sequential Modeling for Molecular 3D Conformation Generation

## Abstract

In the realms of chemistry and drug discovery, the generation of 3D low-energy molecular conformers is critical. While various methods, including deep generative and diffusion-based techniques, have been developed to predict 3D atomic coordinates and molecular geometry elements like bond lengths, angles, and torsion angles, they often neglect the intrinsic correlations among these elements. This oversight, especially regarding torsion angles, can produce less-than-optimal 3D conformers in the context of energy efficiency. Addressing this gap, we introduce a method that explicitly models the dependencies of geometry elements through sequential probability factorization, with a particular focus on optimizing torsion angle correlations. Experimental evaluations on benchmark datasets for molecule conformer generation underscore our approach's superior efficiency and efficacy.

## 1 Introduction

The task of generating 3D molecular conformers centers around producing sets of molecules that exhibit definitive 3D coordinates and are characterized by low-energy conformations. These low-energy conformers are indicative of the molecule's most stable states and are typically the configurations observed in chemical experiments. This significant undertaking serves as a bedrock within the fields of chemistry and drug discovery (Schütt et al., 2018). Furthermore, the 3D structure of a molecule is of paramount importance, with its influence resonating deeply across both the biological and chemical realms (Thomas et al., 2018; Gasteiger et al., 2020; Gebauer et al., 2022; Jing et al., 2021; Batzner et al., 2022; Liu et al., 2021; Geiger & Smidt, 2022).

Generating 3D molecular conformers presents two critical challenges: achieving low-energy states and ensuring efficient generation. The quest for low-energy states is paramount, as these are not only the most stable configurations of a molecule but also the most biologically and chemically relevant. Achieving these states directly influences the accuracy and relevance of subsequent experimental or computational endeavors (Rappe et al., 1992; Halgren, 1996). Parallelly, generation efficiency is crucial. In practical applications, especially in high-throughput scenarios common in drug discovery and chemical analysis, the ability to rapidly and efficiently generate a multitude of conformers can be the difference between breakthrough and bottleneck (Ton et al., 2020; Bilodeau et al., 2022). Striking the right balance between these two imperatives — the precision of low-energy states and the speed of efficient generation — defines the intricacy and importance of the 3D molecular conformers generation task.

Numerous machine learning strategies have emerged to address the intricate task of 3D molecular conformer generation. Among these, MPNN (Gilmer et al., 2017; Yang et al., 2019), a notable graph convolution network, adeptly updates the features of atom nodes and bond edges, leveraging bond edges for message passing and making predictions for coordinates of atoms. However, early applications of such approaches (Simm & Hernández-Lobato, 2020; Xu et al., 2021; Shi et al., 2021; Luo et al., 2021) yielded results that lagged behind those of OMEGA (Hawkins et al., 2010), a leading cheminformatic method. These methods' primary challenge is the expansive search space, often too vast to pinpoint optimal solutions. GeoMol (Ganea et al., 2021) introduced a strategy that focuses on local geometry information—such as bond length, bond angle, and torsion angle—for molecule assembly. This approach effectively narrows the task's search space, sidestepping unnecessary degrees of freedom. More recently, diffusion techniques like GeoDiff (Xu et al., 2022) and Torsional

Figure 1: Runtime vs Performance.

Diffusion (Jing et al., 2022) have emerged, setting new benchmarks in performance and even surpassing OMEGA. Nonetheless, while these diffusion-centric models bring enhanced accuracy, they grapple with substantial computational demands, leading to efficiency challenges.

In light of torsion angles' pivotal role in determining 3D conformer geometry, our study emphasizes torsion angle generation, drawing inspiration from (Jing et al., 2022). Recognizing the intricate interplay between torsion angles within conformers, we advocate for a method that explicitly models these dependencies using probability factorization. This representation is then modeled through recurrent neural architectures, including LSTM (Hochreiter & Schmidhuber, 1997), Bidirectional LSTM (Schuster & Paliwal, 1997), and GRU (Dey & Salem, 2017). We introduce a strategy incorporating a random torsion angle deviation to address the gradient vanishing challenge inherent to angle predictions made with the $\tanh(\cdot)$ function. This ensures the target torsion angle is uniformly distributed within the range $[-\pi, +\pi]$, enhancing model training efficacy.

Building on these foundational ideas, we present TorSeq, a sequential model tailored for torsion angle prediction. This architecture harnesses the power of an MPNN for graph feature encoding, paired with an LSTM for sequential torsion angle forecasts. Preliminary findings, as depicted in Fig 1, underscore our model's prowess. Notably, TorSeq emerges as the inaugural non-diffusion method to surpass the benchmark set by the leading cheminformatics tool, OMEGA(Hawkins et al., 2010)(Hawkins & Nicholls, 2012), all the while boasting commendable runtime efficiency. When compared to the apex diffusion method, our model stands out, delivering not only swifter computations but also rivaling its geometric precision and chemical property outcomes. Moreover, fusing TorSeq with existing diffusion models propels performance to new heights, setting a fresh industry standard. The key contributions of our paper can be encapsulated as follows:

• TorSeq is the pioneering machine-learning approach to introduce an artificial torsional sequence, enabling explicitly modeling interrelations among torsion angles.
• A random torsion angle deviation approach is proposed to overcome the gradient vanishing issue.
• The experimental results demonstrate the effectiveness and efficiency of the proposed methods.

## 2 RELATED WORK

**GeoDiff** (Xu et al., 2022) is the first diffusion method that changes the search space from $\mathbb{R}^{n \times n}$ to $\mathbb{R}^{3 \times n}$. This allows for the 3D coordinates of the atoms in each step to be readily available, thereby enabling the implementation of equivariant graph convolution layers. Although this improvement enhances performance, the atom coordination still experiences redundant degrees of freedom. As a result, GeoDiff requires thousands of denoising steps to achieve optimal functioning.

**GeoMol** (Ganea et al., 2021) introduced a novel approach to derive 3D geometry information, focusing on geometric parameters such as bond length, bond angle, and torsion angle, rather than relying on atoms. This method reduces redundant degrees of freedom, thereby narrowing the search space for the task. A molecule graph is adequate for predicting local features because significant energetic factors tightly constrain the distributions of bond lengths and angles. However, GeoMol faces challenges with torsion angle prediction. Due to the inherent ambiguity in defining arbitrary torsion angles, GeoMol's accuracy in this aspect is limited.

**Torsional Diffusion** (Jing et al., 2022) further narrows the search space to the torsion angle of rotatable bonds only, using RDKit to quickly generate local structure. The search space has been significantly reduced, resulting in fewer denoising steps and outperforming existing methods. However, as a diffusion method, it still requires a few steps to find the best solution. Therefore, Torsional Diffusion still has a tens of times slower runtime than the cheminformatics method.

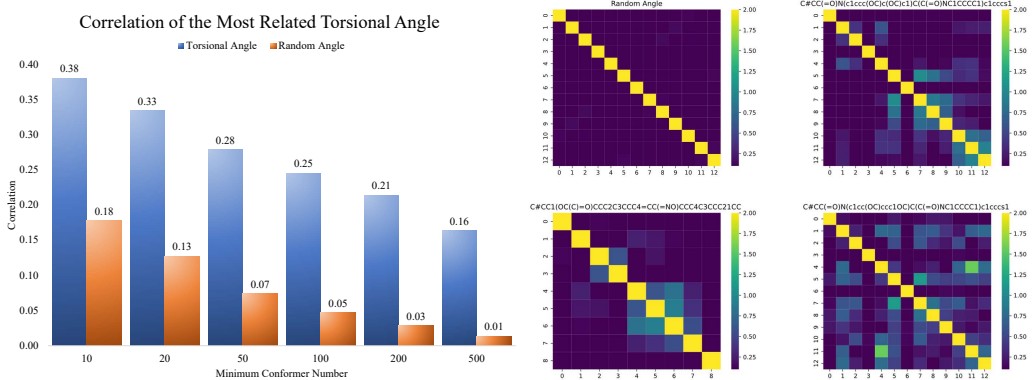

Figure 2: Illustration of correlation among torsion angles. **Left**: Mean of correlations of torsion angles in each molecule; **Right**: Correlation heatmap among torsion angles of some randomly selected molecules.

## 3    TORSEQ: A TORSION SEQUENTIAL MODELING APPROACH

We propose a torsion sequential modeling approach for molecular 3D conformation generation.

### 3.1    MOTIVATION AND CHALLENGE

The task of generating molecular 3D conformations focuses on producing conformer geometry information that remains invariant under $SE(3)$ transformations, i.e., $SE(3)$-invariance (Fuchs et al., 2020). Current methodologies derive this geometry information from the molecular graph using Graph Neural Networks. It is important to note that different geometric components contribute variably to the energy and 3D structure of the generated conformers (Jing et al., 2022). The cutting-edge approach in the field has streamlined this task, emphasizing the prediction of torsion angles due to their significant influence on the 3D layout (Jing et al., 2022).

In this revised context, the task is defined as follows: For a given molecular graph, represented as $\mathcal{G}$, with $m$ torsion angles denoted $\mathbb{T}$, these torsion angles are modeled in a joint manner as

$$p(\mathbb{T}|\mathcal{G}) = p(\tau_1, \ldots, \tau_m|\mathcal{G}), \qquad (1)$$

where each $\tau_i \in \mathbb{T}$. The interrelations between the torsion angles are implicitly captured through graph-based feature encoding. However, this form of joint modeling doesn't adequately represent the dependencies between torsion angles. We argue that to generate a molecular 3D layout with low energy, the inter-dependencies among torsion angles need to be considered, particularly for adjacent ones. To bolster this claim, we investigate the correlations among torsion angles in low-energy conformers.

To this end, we utilize the benchmark dataset GEOM-DRUGS (Axelrod & Gomez-Bombarelli, 2022). We focus on molecules with a minimum of ten conformers to minimize noise. Recognizing the periodic nature of angles, we employ the circular-circular correlation coefficient for measuring the correlations (Mardia & Jupp, 1999). We extract torsion angles using the approach proposed in Section 3.2. To underscore the correlations present among torsion angles, we juxtapose the median correlation of arbitrary angles against that of torsion angles within conformers. As depicted in Figure 2 (left), there's a marked distinction between the correlations of random angles and torsion angles. This underscores the pronounced relationships among torsion angles, particularly the adjacent ones. Meanwhile, Figure 2 (right) illustrates the correlations for torsion angles in a set of randomly selected molecules. The statistics garnered from these observations show that there are strong correlations between torsion angles, which supports the necessity of explicitly modeling the interrelations among torsion angles.

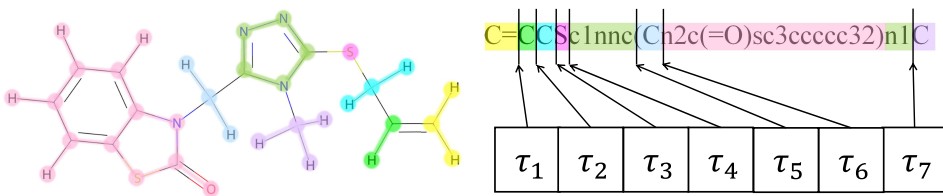

Figure 3: Molecule Graph to Torsional Sequence, **left**: Split Molecule by rotatable bonds; **right**: Get the sequence of rotatable bonds (torsion angles) from SMILES.

## 3.2 EXPLICIT MODELING OF TORSION ANGLE INTERRELATIONS

In this section, we introduce a methodology to capture the intricate interrelationships among torsion angles. Drawing inspiration from conditional language models, which allocate probabilities to word sequences contingent upon a given context, we employ a similar principle for torsion angle modeling. Specifically, we utilize the chain rule of probability to decompose the joint probability represented in Eq. (1) into a series of conditional probabilities, as expressed below:

$$p(\mathbb{T}|\mathcal{G}) = p(\tau_1, \ldots, \tau_m|\mathcal{G}) = \prod_{t=1}^{m} p(\tau_t|\mathcal{G}, \tau_1, \tau_2, \ldots, \tau_{t-1}). \tag{2}$$

From this decomposition, each torsion angle $\tau_t$ is predicted based on $\mathcal{G}, \tau_1, \tau_2, \ldots$, and $\tau_{t-1}$. Consequently, our primary focus narrows down to characterizing the probability distribution of the next torsion angle, thereby allowing us to systematically integrate dependencies between torsion angles.

A widely-adopted approach for modeling these conditional probabilities leverages recurrent neural networks (RNNs) (Mikolov et al., 2010), particularly architectures like the Long Short Term Memory (LSTM) networks (Hochreiter & Schmidhuber, 1997; Schuster & Paliwal, 1997). The appeal of RNNs lies in their capacity to avoid making Markov assumptions, thereby avoiding conditional independence constraints. In this work, we employ LSTM to capture the conditional probabilities associated with torsion angles.

**Torsion sequence definition.** In the training and inference processes of a recurrent neural network, it's imperative to first establish the sequential dependencies among torsion angles. Yet, drug-like molecules often possess rings and irregular branches, which lack a clear sequential structure. This makes it particularly challenging to derive a sequence of torsion angles from the molecule without compromising its structural integrity. To address this challenge, we employ string-based molecular representations, such as the Simplified Molecular-Input Line-Entry System (SMILES) (Weininger, 1988). The atomic order in a SMILES string provides a linear description of the molecule's structure. Utilizing molecular canonicalization techniques (Weininger et al., 1989), we ensure that every unique molecule corresponds to a singular SMILES string. With the canonical string of a molecule and its associated rotatable bonds with torsion angles in hand, we then arrange the torsion angles according to the positions of their respective rotatable bonds. This approach allows us to determine a definitive sequence for the torsion angles within a molecule. This process is illustrated in Figure 3.

**Canonical-based torsion angle definition.** Next, we address the ambiguity inherent in defining torsion angles (Ganea et al., 2021; Jing et al., 2022). Essentially, a torsion angle is determined by a rotatable bond between atoms and an arbitrary selection of neighboring atoms. Due to this arbitrary selection, the angle associated with the same rotatable bond may differ based on the neighboring atoms chosen, resulting in an angle that lacks clarity, reliability, and consistent contextual meaning.

To rectify this ambiguity, we introduce a method that anchors the definition of a torsion angle to the ranked neighbors of the two terminal atoms of the rotatable bond. To illustrate, let's consider a rotatable bond, $e(i, j)$, with $i$ and $j$ representing its two terminal atoms. Initially, we rank the sets $\mathcal{N}_i - \{j\}$ and $\mathcal{N}_j - \{i\}$ according to atom positions within the canonical string representation of the molecule, where $\mathcal{N}_i$ and $\mathcal{N}_j$ denote the neighboring atoms of $i$ and $j$, respectively. From these ranked sets, we then select the atoms with the lowest position indices, termed $l$ from $\mathcal{N}_i$ and $k$ from $\mathcal{N}_j$. The torsion angle of $e(i, j)$ is subsequently defined as the angle delineated by the intersection of the planes formed by atoms $l, i$, and $j$ and by atoms $i, j$, and $k$.

### 3.3 INCORPORATING RANDOM ANGLES DEVIATION IN TORSION ANGLE PREDICTION

Traditionally, the $\tanh(\cdot)$ function is employed to restrict outputs within the interval $[-1, +1]$. This ensures that the outputs can be conveniently mapped to the range $[-\pi, +\pi]$ for angle prediction. Specifically, a torsion angle can be approximated as $\hat{\tau} = \pi \times \tanh(\boldsymbol{h}_\tau)$, where $\boldsymbol{h}_\tau$ denotes the feature representation of $\tau$. Challenges arise when the desired angle approaches $\pm\pi$. Near these extremities, the gradient of $\tanh(\cdot)$ tends towards zero, inducing the well-known vanishing gradient issue. This makes it difficult for models to accurately learn target torsion angles near $\pm\pi$. This challenge is evident in figure 4, illustrating the torsion angle distribution of a molecule: torsion angles situated near $-\pi$ present significant learning difficulties for the model.

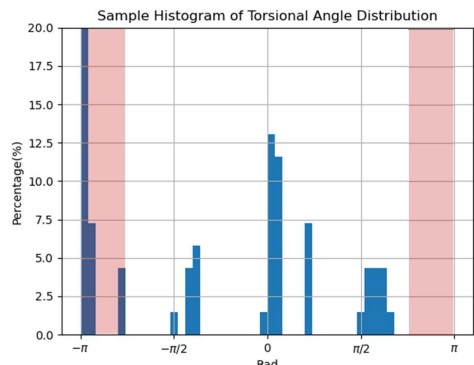

Figure 4: Torsion angle distribution in a GEOM-DRUGS molecule's conformers.

The core challenge arises when the model attempts to predict angles situated within regions where the gradient of $\tanh(\cdot)$ approaches zero. To tackle this issue, we introduce an innovative approach by incorporating a random angle deviation selected from the interval $[-\pi, \pi]$. The loss in this context is determined by loss $= f(\hat{\tau}', \tau')$, where $f(\cdot)$ is a loss function, $\tau'$ is defined as $\tau' = (\tau - \Delta\tau + \pi)$ mod $2\pi - \pi$ and $\Delta\tau$, taken from the range $[-\pi, \pi]$, represents a random angular deviation. By this design, the target angle is adaptively shifted, ensuring its potential position spans anywhere between $-\pi$ and $\pi$. This strategy effectively mitigates the vanishing gradient issue, facilitating more efficient model training. During the inference of a torsion angle, a random angle deviation is generated and fed into the model, combined with the graph and previously generated torsion angles. The resulting predicted angle is computed by adding the model's output to this angle difference $\hat{\tau} = \hat{\tau}' + \Delta\tau$.

### 3.4 MODEL ARCHITECTURE

This section introduces our proposed Torsion Sequential model (TorSeq).

**Feature encoding.** Given an input $\mathcal{G} = (\mathcal{V}, \mathcal{E})$, $u \in \mathcal{V}$ and $e \in \mathcal{E}$ denote a node and an edge, respectively. We employ a Message-Passing Neural Network (MPNN) to encode node and edge features. At the $t^{th}$ layer, node $u$'s features $\boldsymbol{h}_u$ and edge $e_{(u,v)}$'s features $\boldsymbol{e}_{(u,v)}$ are updated as

$$\boldsymbol{e}_{(u,v)}^{t+1} = MLP_1(\boldsymbol{h}_u^t + \boldsymbol{h}_v^t + \boldsymbol{e}_{(u,v)}^t) + (1 + \boldsymbol{\phi})\boldsymbol{e}_{(u,v)}^t, \tag{3}$$

$$\boldsymbol{h}_u^{t+1} = (1 + \boldsymbol{\psi})\boldsymbol{h}_u^t + MLP_2(\sum_{v \in \mathcal{N}_u} MLP_3(\boldsymbol{e}_{(u,v)}^{t+1})), \tag{4}$$

where $MLP_1$, $MLP_2$ and $MLP_3$ are multi-layer perception layers, and $\boldsymbol{\phi}$ and $\boldsymbol{\psi}$ are learnable parameters. Based on node-level and edge-level features, we compute a motif's feature by aggregating the features of nodes in it. Then, we follow (Zhang et al., 2021) to use a graph convolutional network to encode the motif-level features. Finally, by aggregating the node and motif features, we compute the molecule level feature $\boldsymbol{h}_{mol}$. Also, for the rotatable bond $(u, v)$, to learn the local structure from both sides, we aggregate the feature of neighbor $h_{neighbor}$; the neighbor feature of node $u$ is:

$$\boldsymbol{n}_u = MLP(\sum_{w \in \mathcal{N}_u, w \neq v} \boldsymbol{h}_u) \tag{5}$$

**Torsion angle features.** For each torsion angle $\tau_{i,j,k,l}$, we use following features, node features $\boldsymbol{h}_i, \boldsymbol{h}_j, \boldsymbol{h}_k, \boldsymbol{h}_l$, edge feature $\boldsymbol{e}_{(i,j)}, \boldsymbol{e}_{(j,k)}, \boldsymbol{e}_{(k,l)}$, motif feature $\boldsymbol{m}_j, \boldsymbol{m}_k$, graph feature $h_\mathcal{G}$, neighbor feature $n_j, n_k$. In addition, we add the random start embedding $\boldsymbol{h}_{\Delta\tau}$. All these features are concatenated as the torsion angle feature $\boldsymbol{h}_\tau$.

**Sequential torsion prediction head.** Based on the features of torsion angles, we choose to use a bidirectional LSTM (Schuster & Paliwal, 1997) to learn the explicit dependencies of torsion angles. The final prediction for a torsion angle is $\hat{\tau} = \tanh(\boldsymbol{h}_\tau) + \Delta\tau$, $\Delta\tau$ is the random torsion angle deviation. Our approach employs changing torsion angles by following the same approach as (Jing et al., 2022), which consequently preserves the $SE(3)$ invariance property.

**Loss function.** Our loss function comprises two components: the torsion angle loss and the random angle loss. The torsion angle loss is determined by comparing the predicted torsion angles to the ground truth values. For a given molecule that has $N$ ground truth conformers, TorSeq will generate $M$ conformers. Following the approach of (Ganea et al., 2021), we employ the Earth Mover's Distance loss (Flamary et al., 2021) to ensure that the model's predicted torsion angles closely align with the target ground truth conformer. The random angle loss is computed based on the difference $\Delta\tau$ and its predicted counterpart $\hat{\Delta}\tau$. Specifically, for the set of predicted torsion angles of conformer $m$ and the ground truth torsion angle set of conformer $n$, the loss is calculated as

$$\mathcal{L} = \frac{1}{|\mathbb{T}|} \sum_{i \in |\mathbb{T}|} 1 - \cos(\hat{\Delta}\tau_i - \Delta\tau_i) + \sum_{m \in M, n \in N} W_{m,n} \frac{1}{|\mathbb{T}|} \sum_{\hat{\tau}_i \in \hat{\mathbb{T}}_m, \tau_i \in \mathbb{T}_n} 1 - \cos(\hat{\tau}_i - \tau_i) \quad (6)$$

where $W$ denotes the weight calculated by the Earth Movement Distance (EMD). The first term is the random angle loss and the second term is the torsion angle loss.

### 3.5 HARMONIZING TORSEQ WITH TORSIONAL DIFFUSION

The diffusion model has been effectively employed for generating 3D molecular conformers (Jing et al., 2022). Due to efficiency considerations, TorSeq does not directly incorporate the diffusion model. Nevertheless, our proposed approach is complementary to the diffusion model, and the two can be seamlessly integrated. Specifically, we leverage the Torsional Diffusion model and replace torsion prediction component with TorSeq's sequential modeling methodology. Through this modification, torsion angles are sequentially predicted within the revised diffusion framework, which we designate as Tor.Diff + TorSeq.

## 4 EXPERIMENTAL STUDY

We evaluate TorSeq on the low-energy conformer generation task.

### 4.1 EVALUATION SETUP

**Dataset.** We evaluate TorSeq on two benchmark datasets: GEOM-DRUGS (Axelrod & Gomez-Bombarelli, 2022) and GEOM-QM9 (Ramakrishnan et al., 2014). We follow the same train/val/test split and pre-processing strategy as described in (Ganea et al., 2021) and (Jing et al., 2022).

**Metric.** To evaluate the geometric structure, we use Average Minimum RMSD (AMR) and Coverage. Both metrics report recall and precision. For a molecule with $N$ ground-truth conformers, we generate $M$ conformers ($M = 2N$), and the coverage and AMR of recall and precision are calculated as

$$\text{COV-R} = \frac{1}{N} \left| \{ n \in [1...N] : \exists m \in [1...M], RMSD(C_n, \hat{C}_m) < \delta \} \right|, \quad (7)$$

$$\text{AMR-R} = \frac{1}{N} \sum_{n \in [1...N]} \min_{m \in [1...M]} RMSD(C_n, \hat{C}_m), \quad (8)$$

$$\text{COV-P} = \frac{1}{M} \left| \{ m \in [1...M] : \exists n \in [1...N], RMSD(C_n, \hat{C}_m) < \delta \} \right|, \quad (9)$$

$$\text{AMR-P} = \frac{1}{M} \sum_{m \in [1...M]} \min_{n \in [1...N]} RMSD(C_n, \hat{C}_m). \quad (10)$$

Essentially, recall measures finding the best-fit generated conformer for each ground-truth conformer, focusing more on diversity. Precision, on the other hand, measures the quality of generated conformers. We also measure the running time and chemical properties as (Jing et al., 2022).

**Baseline models.** We benchmark our approach against current state-of-the-art models. We assess cheminformatics techniques such as RDKit ETKDG (Riniker & Landrum, 2015) and OMEGA (Hawkins et al., 2010), (Hawkins & Nicholls, 2012). Additionally, we evaluate machine learning methodologies, considering both non-diffusion methods like GeoMol (Ganea et al., 2021) and diffusion-centric techniques, specifically GeoDiff (Xu et al., 2022) and Torsional Diffusion (Jing et al., 2022)

Table 1: Result on GEOM-DRUGS Dataset, without FF optimization. The Coverage (%) is based on threshold $\delta = 0.75$ Å.

| | Recall | | | | Precision | | | |
|---|---|---|---|---|---|---|---|---|
| | Coverage ↑ | | AMR ↓ | | Coverage ↑ | | AMR ↓ | |
| Method | Mean | Med | Mean | Med | Mean | Med | Mean | Med |
| OMEGA | 53.4 | 54.6 | 0.841 | 0.762 | 40.5 | 33.3 | 0.946 | 0.854 |
| RDKit ETKDG | 38.4 | 28.6 | 1.058 | 1.002 | 40.9 | 30.8 | 0.995 | 0.895 |
| GeoMol | 44.6 | 41.4 | 0.875 | 0.834 | 43.0 | 36.4 | 0.928 | 0.841 |
| **TorSeq (ours)** | **55.5** | **56.2** | **0.773** | **0.748** | **52.6** | **53.8** | **0.806** | **0.744** |
| GeoDiff | 42.1 | 37.8 | 0.835 | 0.809 | 24.9 | 14.5 | 1.136 | 1.090 |
| Torsional Diffusion | 72.7 | 80.0 | 0.582 | 0.565 | 55.2 | 56.9 | 0.778 | 0.729 |
| Tor.Diff+**TorSeq(ours)** | **72.8** | **80.6** | **0.580** | **0.558** | **55.9** | **58.6** | **0.769** | **0.725** |

* The performance of baseline methods are borrowed from (Jing et al., 2022), and we use. the same dataset and split as (Ganea et al., 2021) (Jing et al., 2022).

Table 2: Runtime (second per conformer) and Median AMR.

| Method | Steps | Runtime | AMR-R | AMR-P |
|---|---|---|---|---|
| RDKit ETKDG | - | **0.05** | 1.002 | 0.895 |
| OMEGA | - | 0.14 | 0.762 | 0.854 |
| GeoMol | - | 0.19 | 0.834 | 0.841 |
| Torsional Diffusion | 5 | 2.43 | 0.685 | 0.963 |
| Torsional Diffusion | 10 | 4.03 | 0.580 | 0.791 |
| Torsional Diffusion | 20 | 7.63 | **0.565** | **0.729** |
| GeoDiff | 5000 | 505.97 | 0.809 | 1.090 |
| **TorSeq (ours)** | - | **0.13** | 0.748 | 0.744 |

## 4.2 ENSEMBLE GEOMETRIC RESULTS

We initiated our evaluation using the GEOM-DRUGS dataset for TorSeq. The performance metrics are presented in Table 1. Relative to non-diffusion models, TorSeq exhibits superior performance. Compared to the GeoMol and GeoDiff, two advanced machine-learning methods proposed in recent years, our method reduces the average minimum RMSD recall by 8% and the average minimum RMSD precision by 12%. At the same time, our method performs better than the state-of-the-art cheminformatic OMEGA in both performance and running speed. When juxtaposed with diffusion-based models, our integrated Tor.Diff+TorSeq model also outshines the Torsional Diffusion model in every metric, with an advantage of 1.3% in median AMR-R and 0.4% in median AMR-P. These results underscore the efficacy of our proposed TorSeq.

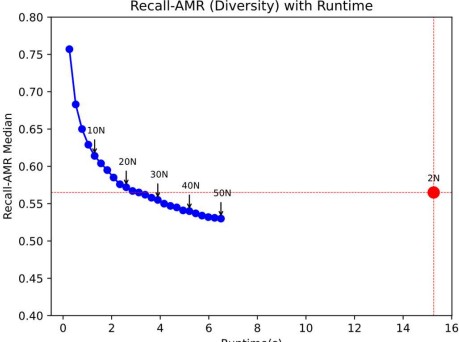
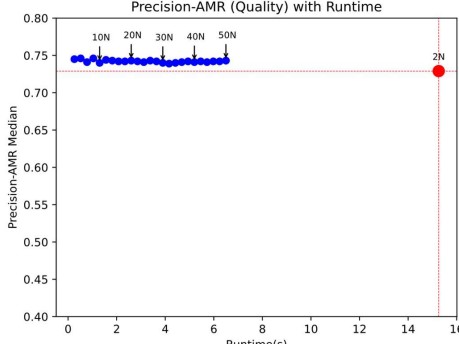

Figure 5: Runtime vs Perfomance **left**: Recall and Runtime **right**: Precision and Runtime. The blue points are performances of TorSeq, while the red points are performances of the TorDiff model. $N$ is the number of ground truth conformers. $KN$ means generating $K$ times of conformers.

Table 3: Comparison results of chemical properties. We report the generated Conformers Boltzmann-weighted Chemical Properties including $E$, $\Delta\epsilon$ and $E_{min}$ in kcal/mol, $\mu$ in debye.

| Method | $E$ | $\mu$ | $\Delta\epsilon$ | $E_{min}$ |
|---|---|---|---|---|
| RDKit ETKDG | 0.81 | 0.52 | 0.75 | 1.16 |
| OMEGA | 0.68 | 0.66 | 0.68 | 0.69 |
| GeoMol | 0.42 | 0.34 | 0.59 | 0.40 |
| GeoDiff | 0.31 | 0.35 | 0.89 | 0.39 |
| Torsional Diffusion | **0.22** | 0.35 | 0.54 | **0.13** |
| **TorSeq (ours)** | 0.24 | **0.29** | **0.40** | **0.13** |

\* The performance of baseline methods are from (Jing et al., 2022). We use the same set of molecules as (Jing et al., 2022).

## 4.3 RUNTIME ANALYSIS

In this section, we conduct an efficiency analysis to assess the performance of our proposed methods. Under the same hardware configurations, we evaluate the runtime (in seconds) required by TorSeq and the baseline models to generate a single conformer. All evaluations are performed on a CPU (AMD Ryzen 7 3800X 8-Core Processor). In alignment with (Jing et al., 2022), we allocate eight threads and randomly select 10 molecules, generating eight conformers for each. The comparative runtime results are presented in Table 2.

From the data in Table 2, our approach boasts a pronounced efficiency advantage over other machine learning models. Specifically, when Torsional Diff undergoes a 20-step denoising process, our method operates 50 times faster. Remarkably, even when TorDiff is set to denoise in just ten steps, TorSeq still exhibits a speed more than 20 times that of TorDiff while achieving a 6% improvement in median AMR-P.

This enhanced efficiency enables TorSeq to produce a greater number of conformers within the same timeframe, potentially enriching the diversity of the generated structures. To validate this hypothesis, we scaled up the number of generated conformers and documented the subsequent evaluation outcomes in Fig 5. The results revealed a consistent improvement in AMR-Recall performance, while the AMR-Precision remained stable. Leveraging its efficiency in generation, TorSeq can surpass the recall performance of TorDiff by producing more conformers in significantly less time.

## 4.4 CHEMISTRY PROPERTY RESULTS

We subsequently assess the chemical properties of the generated conformers. In line with Torsional Diffusion (Jing et al., 2022), we utilize the same 100 molecules from the test set, producing $min(2N, 32)$ conformers for each molecule. Before evaluating these conformers' chemical attributes, we first stabilize them using the GFN2-xTB software (Bannwarth et al., 2019). Following this, the xTB software is employed to gauge properties including energy $E$, dipole moment $\mu$, HOMO-LUMO gap $\Delta\epsilon$, and the minimum energy $E_{min}$. The comparative outcomes are detailed in Table 3. Notably, our methodology surpasses Torsional Diffusion in both dipole moment and HOMO-LUMO metrics, while delivering comparable results in energy and minimum energy.

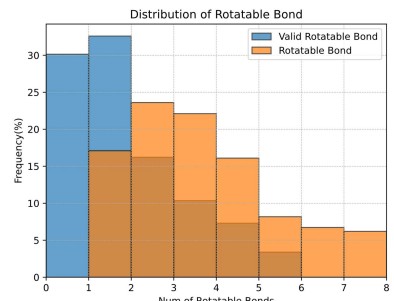

Figure 6: The distribution of rotatable bond and valid rotatable bond.

## 4.5 PERFORMANCE ON SMALL MOLECULE

In this section, we evaluate our proposed methods using an alternative benchmark dataset: GEOM-QM9. This dataset primarily comprises smaller molecules when contrasted with GEOM-DRUGS. Under identical settings, we juxtapose the performance of our model against established baseline models. The results can be perused in Table 4. As evident, among cutting-edge deep learning approaches, TorSeq's efficacy on smaller molecules is surpassed only by Torsional Diffusion Jing et al.

Table 4: Result on GEOM-QM9 Dataset, without FF optimization. The Coverage (%) is based on threshold $\delta = 0.5$ Å

| | Recall | | | | Precision | | | |
|---|---|---|---|---|---|---|---|---|
| | Coverage ↑ | | AMR ↓ | | Coverage ↑ | | AMR ↓ | |
| Method | Mean | Med | Mean | Med | Mean | Med | Mean | Med |
| RDKit ETKDG | 85.1 | **100** | 0.235 | 0.199 | 86.8 | **100** | 0.232 | 0.205 |
| OMEGA | 85.5 | **100** | 0.177 | **0.126** | 82.9 | **100** | 0.224 | **0.186** |
| GeoMol | 91.5 | **100** | 0.225 | 0.193 | 86.7 | **100** | 0.270 | 0.241 |
| **TorSeq (ours)** | 92.5 | **100** | 0.219 | 0.182 | 89.9 | **100** | 0.244 | 0.215 |
| GeoDiff | 76.5 | **100** | 0.297 | 0.229 | 50.0 | 33.5 | 0.524 | 0.510 |
| Torsional Diffusion | 92.8 | **100** | 0.178 | 0.147 | **92.7** | **100** | 0.221 | 0.195 |
| Tor.Diff+**TorSeq(ours)** | **95.0** | **100** | **0.176** | 0.146 | 91.6 | **100** | **0.220** | 0.194 |

[*] The performance of baseline methods are borrowed from (Jing et al., 2022), and we use. the same dataset and split as (Ganea et al., 2021) (Jing et al., 2022).

(2022). When integrated with Torsional Diffusion, the combined performance of TorSeq+Torsional Diffusion marginally edges out Torsional Diffusion alone. Nonetheless, while the superiority of TorSeq is discernible with the DRUGS dataset, it doesn't shine as prominently on the GEOM-QM9 dataset. A potential reason is the high prevalence of molecules in the GEOM-QM9 dataset with a solitary torsion angle. As illustrated in Fig 6, over $60\%$ of molecules in the GEOM-QM9 dataset possess a maximum of one valid rotatable bond. This suggests that our TorSeq model might face challenges in achieving optimal performance on smaller molecules, particularly when constrained by the torsion angle sequence length.

## 4.6 ABLATION STUDY

In this section, we explore the impact of each introduced component on TorSeq's overall performance. Specifically, we evaluate the model after individually omitting the proposed conditional model component, the LSTM Block, and the random torsion angle deviation. We base our evaluations on the GEOM-DRUGS dataset and focus on metrics such as median AMR-Recall and median AMR-Precision. The results from this ablation study are detailed in Table 5.

Table 5: Ablation Study of TorSeq.

| Method | AMR-R | AMR-P |
|---|---|---|
| TorSeq | 0.748 | 0.744 |
| (-) LSTM Block | 0.778 | 0.786 |
| (-) Random Deviation | 0.773 | 0.782 |

A comparative analysis of results with and without the LSTM block reveals its significant influence: there's a notable improvement in recall and precision by 3% and 4.2%, respectively. Similarly, excluding the random torsion angle deviation. results in performance dips of 2.5% and 3.8% in recall and precision, respectively. This underscores the utility of the random torsion deviation method in addressing the vanishing gradient challenge.

## 5 CONCLUSION

In this study, we introduce TorSeq, a novel approach to 3D molecular conformer generation. To effectively encapsulate the inter-dependencies among torsion angles, we factorize the joint probability into a sequence of conditional probabilities, thereby explicitly accounting for the inherent dependencies during torsion angle generation. Addressing the vanishing gradient challenge often encountered with the usage of $\tanh(\cdot)$ for angle predictions, we innovatively incorporate a random angle deviation. This strategy ensures the target angle can potentially shift to regions with more pronounced gradients. Empirical evaluations across two benchmark datasets underline the potency and robustness of our methodology. Moreover, TorSeq's marked efficiency over diffusion models paves the way for broader practical applications in the realm of molecular modeling.

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

## A    VISUALIZATION

We utilize the molecules chosen by (Ganea et al., 2021) to evaluate the conformers generated by TorSeq. This selection is based on the most recent paper in this field, notable for its comprehensive visual comparison. The baseline cases are documented in (Ganea et al., 2021) Appendix, specifically in Fig.13.

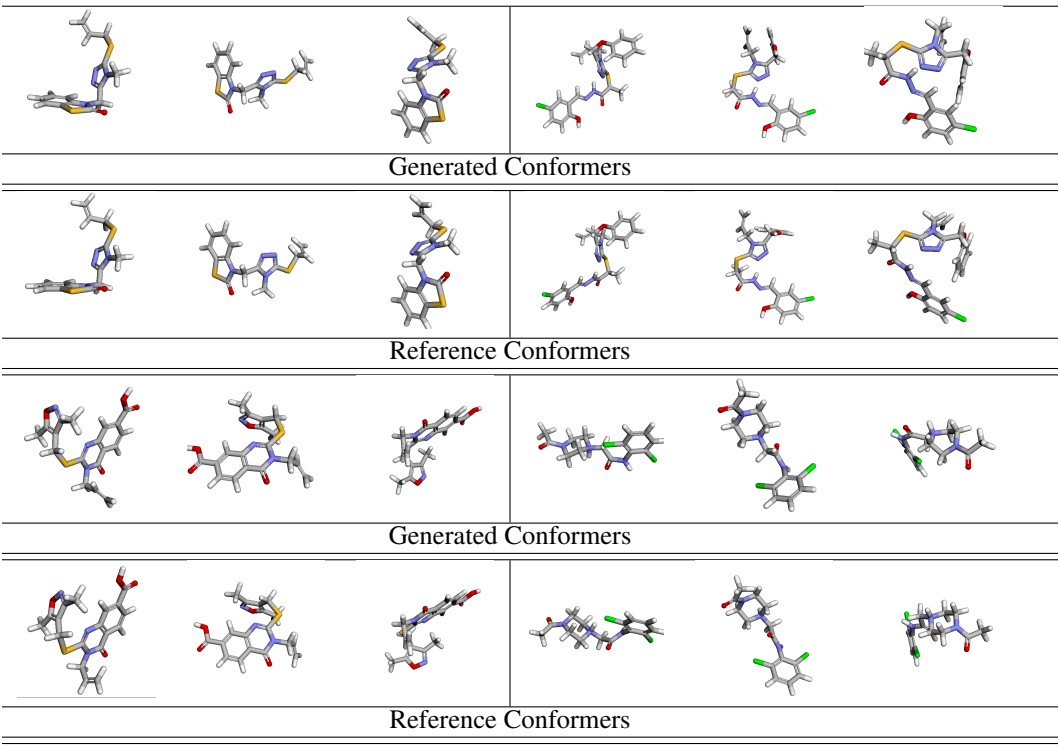

Generated Conformers

Reference Conformers

Generated Conformers

Reference Conformers

Figure 7: Steric Clashs on GEOM-DRGUS Dataset

## B    ANALYSIS IN STERIC CLASHES

### B.1    NUMERICAL STERIC CLASHE ANALYSIS

At the same time, Torsional Diffusion (Jing et al., 2022) represents the current state-of-the-art model, and its generated conformers are openly accessible. Therefore, our numerical comparisons are confined to TorSeq, Torsional Diffusion, and the combination of Torsional Diffusion with TorSeq. We utilize all 197.1k conformers generated from the test set of GEOM-DRUGS (Axelrod & Gomez-Bombarelli, 2022) dataset, and in accordance with (Chen et al., 2010; Williams et al., 2018), we adopt a threshold of $\delta = 0.4\text{\AA}$. We report the rate of conformers exhibiting steric clashes, the proportion without steric issues to highlight how many generated conformers encounter this problem, and the average number of problematic atom pairs to illustrate the severity of steric clashes in adverse cases.

Table 6: Steric Clashes Result

| Method | Without Steric Clashes | With Steric Clashes | Bad atom pairs per Error |
|---|---|---|---|
| **TorSeq** | 98.899 % | 1.101% | 1.11 |
| Tor. Diff | 99.951% | 0.049% | 1.06 |
| Tor. Diff + **TorSeq** | **99.964%** | **0.036%** | **1.03** |

From Table. 6, we can find that all three methods can treat steric clashes as a kind of worst case in generated conformers. The significant improvement in reducing the worst-case indicator for the

latter two models is attributed to the equivariant model's ability to obtain atomic coordinates. Consequently, compared to TorSeq, the MPNN the backbone model is the reason that fails to capture molecular coordinates. Equivariant models demonstrate substantial advantages in lowering worst-case rates.

At the same time, the combination of TorSeq with Torsional Diffusion further reduced the error rate by 25%. This outcome underscores the vital role of sequential modeling in the synergistic use of equivariant models and diffusion models.

The third metric reveals that the extent of steric clash in the three models is not severe, typically involving only a pair of atomic groups.

## C  DISCUSSION

In the task of Molecular Conformation Generation (MCG), sequential modeling is a crucial complement. Molecular modeling is usually limited to 3 to 4 GNN layers, which is insufficient for comprehending multiple-hop topological relationships.

