# OpenReview forum: "TorSeq: Torsion Sequential Modeling for Molecular 3D Conformation Generation"
_ICLR.cc/2024/Conference — Submitted to ICLR 2024_

### Official Review · Reviewer_Mhor · 2023-10-29

**Soundness:** 2 fair
**Presentation:** 3 good
**Contribution:** 2 fair
**Rating:** 3
**Confidence:** 4

**Summary:**

This paper introduces TorSeq, a new method for 3D molecular conformation generation that focuses on predicting torsion angles between atoms using sequential probability factorization and LSTM networks. A key innovation is the modeling of dependencies between torsions. The method also incorporates a novel random torsion angle deviation during training to avoid vanishing gradients. Experiments demonstrate state-of-the-art accuracy and efficiency for TorSeq over existing methods like OMEGA on standard benchmarks.

**Strengths:**

The paper is well-written and presents the impressive runtime of TorSeq, which is indeed remarkably fast. The experiments conducted are comprehensive, featuring evaluations on two benchmark datasets, comparisons to numerous state-of-the-art baselines, and ablation studies. Furthermore, the authors have made the code and datasets publicly available, fostering reproducibility in the field.

**Weaknesses:**

- The related work section appears insufficient, as only three previous studies are mentioned. A more comprehensive review of the literature would strengthen the paper.
- The proposed method does not seem to outperform "Torsional Diffusion" as a standalone technique. While combining it with "Torsional Diffusion" yields better results than "Torsional Diffusion" alone, the increased runtime compromises its efficiency. Thus, the paper's claim of achieving both effectiveness and efficiency is not well-supported, as a trade-off exists between the two.
- A key aspect of TorSeq is defining a suitable sequence of torsions to capture structural dependencies. Relying on SMILES ordering is a straightforward yet imperfect solution, as the model may underperform or fail if torsions are provided in an incorrect or suboptimal order. The authors should analyze the sensitivity of their approach to changes in the torsion sequence more thoroughly.
- The paper focuses on single conformation generation, which is not as relevant as generating Boltzmann distributions, considering that low-energy states form a Boltzmann distribution rather than a single state. Recent works like [1], [2], and [3] have explored Boltzmann distribution generation. The paper's emphasis on maximum likelihood training on the GEOM dataset does not guarantee sampling proportionally to the Boltzmann distribution, making the approach outdated and less interesting.
- The paper should include a more extensive comparison with baseline methods, as only a few are currently examined.

[1] Towards equilibrium molecular conformation generation with GFlowNets.
[2] Towards predicting equilibrium distributions for molecular systems with deep learning.
[3] Boltzmann generators: Sampling equilibrium states of many-body systems with deep learning.

**Questions:**

- How sensitive is the model's performance to variations in the SMILES format or the canonicalization method employed for determining torsion order?
- Can you suggest and assess alternative approaches for defining the torsional sequence, such as using 3D distances? How does the model's accuracy change if torsions are supplied in a random order rather than in the SMILES order?
- Is it possible to enhance the LSTM dependency modeling with positional encodings or attention mechanisms to reduce reliance on sequence order?
- Have you explored any data augmentation techniques during training, like altering the torsion order?
- I recently came across this paper [1] and noted that the performance of simple RDKit + Clustering from [2] is quite impressive. Could you include it as a baseline for comparison?

[1] Towards equilibrium molecular conformation generation with GFlowNets.
[2] Do deep learning methods really perform better in molecular conformation generation?

---

> ### Author Response · Authors · 2023-11-20
>
> Thanks for your valuable review, comments and questions.
>
> ## Weakness 1 Background
>
> Thanks for your suggestion. We will update a more comprehensive background in the final version.
>
> ## Weakness 2 The performance of Torseq(independence) vs previous methods
>
> The COV threshould $\delta = 0.75\AA$, not $1.25\AA$ like most previous work. Note: Chemical Properties use $max (32, 2N_{ref})$ conformers, randomly selected from generated conformers; therefore for TorSeq, the numbers we report in these chemical property metrics are the same.
>
> When considered as an independent technology, TorSeq demonstrates its capability to get better or comparable performance in all metrics listed in [1], and it achieves this with less computational cost.
>
> | Method                 | Prec COV $\uparrow$ | Prec AMR $\downarrow$ | $E$  | $\mu$ | $\Delta \epsilon$ | $E_{min}$ | Recall COV $\uparrow$ | Recall AMR $\downarrow$ | N_{ref} | Average runtime|  Total runtime  |
> |------------------------|:-------------------:|:---------------------:|:----:|:-----:|:-----------------:|:---------:|:---------------------:|:-----------------------:|:-------:|:-------:|:-------:|
> | Rdkit+ETKDG(Chem)      |       33.3 \%       |          0.89         | 0.81 |  0.52 |        0.75       |    1.16   |         28.6\%        |           1.00          |    2    |   0.05  |   0.10  |
> | GeoMol(NeurIPS 2021)   |        36.4\%       |          0.84         | 0.42 |  0.34 |        0.59       |    0.40   |         41.4\%        |           0.83          |    2    |   0.19  |   0.38  |
> | GeoDiff(ICLR 2022)     |        14.5\%       |          1.09         | 0.31 |  0.35 |        0.89       |    0.39   |         37.8\%        |           0.81          |    2    |  505.97 | 1011.94 |
> | Rdkit + Clustering [2] |        26.5\%       |          1.01         |      |       |                   |           |                       |           0.78          |    30   |   0.05  |   1.50  |
> | Tor.Diff(NeurIPS 2022) |        56.9\%       |          0.73         | 0.22 |  0.35 |        0.54       |    0.13   |         80.0\%        |           0.57          |    2    |   7.63  |  15.32  |
> | TorSeq (2N)            |        53.8\%       |          0.74         | 0.24 |  0.29 |        0.40       |    0.13   |         56.2\%        |           0.75          |    2    |   0.13  |   0.26  |
> | TorSeq (10N)           |        54.0\%       |          0.74         | 0.24 |  0.29 |        0.40       |    0.13   |         73.9\%        |           0.61          |    10   |   0.13  |   1.30  |
> | TorSeq (50N)           |        54.4\%       |          0.74         | 0.24 |  0.29 |        0.40       |    0.13   |         84.1\%        |           0.53          |    50   |   0.13  |   6.50  |

---

> ### Author Response · Authors · 2023-11-20
>
> ## Weakness 3, Question 3 and 4, Sequential model and data augmentation:
> In this research, we examine the SMILES, which is organized as a tree structure sorted using the Depth-First Search (DFS) algorithm. In SMILES, numbers indicate rings (which our study does not focus on), and parentheses denote branches. Consequently, we treat the torsion angle sequence similarly, as a tree structure sorted through DFS. Although transitioning from tree structures to sequences inevitably incurs some data loss, our ablation studies have confirmed that sequence modeling, even with this loss, is substantially more effective than any approach neglecting sequence modeling.
>
> Our correlation analysis suggests that DFS is likely more advantageous than Breadth-First Search (BFS) or random sorting methods for organizing torsion angles. We tested two DFS algorithms (first considering smaller branches and default by dataset) to sort these angles, analogous to different canonical SMILES representations. The outcomes varied slightly, by less than 1\%, indicating a minor sensitivity to DFS algorithm variations. Consequently, we initially omitted a detailed discussion on this aspect, but we plan to include these discussions and experimental findings in the final revision of our paper.
>
> The first, named GEOM-DFS, adheres to the default atomic numbering of the dataset. The second strategy gives precedence to shorter branches within the molecular tree. We retrained and tested the model using GEOM-DFS, please see the following table. Our findings indicate that employing different DFS algorithms does not significantly affect model performance
>
> |                     |  |    RMSD AMR $\downarrow$   |           |        |
> |---------------------|:--------:|:------:|:---------:|:------:|
> |                     |  Recall     |    | Precision           |
> | DFS Method          |   Mean   | Median |    Mean   | Median |
> | GEOM-DFS            |  0.768 |  0.745 |   0.814   |  0.758 |
> | Short branch first  |   0.773  |  0.748 |   0.806   |  0.744 |
>
> LSTM is designed to effectively manage long-term information. LSTM achieve this through their unique gating mechanisms, which judiciously determine what information is retained or discarded at each step of a sequence.
> The primary focus of this paper is to explore the importance of sequentially modeling torsion angles. However, we deeply agree that investigating more sophisticated model designs that enhance the sequential modeling of torsion angles presents an exciting avenue for future research.
>
> ## Weakness 4 Boltzmann weight generator:
>
> A key distinction between converting maximum likelihood and Boltzmann distributions lies in the training approach and the reward mechanisms employed, rather than in the modeling process. This distinction is evident in studies such as [1] and [3].
> As a standalone technique, TorSeq, in theory, requires only a modification in the training strategy, such as shifting the target from a uniform distribution to a Boltzmann distribution, to effectively address this issue.
>
> ## Weakness 5 More baseline methods:
>
> Thanks for your suggestions. We include the comparison results with more baseline methods in table below. We will update the table 1 in the final version.
>
> ## Question 5 [2] as baseline:
>
> Thanks for you suggestion. We want to draw your attention that the results reported in [2] are not directly comparable to some SOTA works and ours. This is because there's no chemical properties, precision report in their paper. In this way we cannot make a comprehensive comparison
>
> For fair comparison, we report their results with the same threshold using their github gist: https://gist.github.com/ZhouGengmo. The results are summarized in the first table in our comment.
> From the results, our TorSeq and other SOTA appraoches can outperform [2].
>
> In light of this concern, we recommend a cautious approach to integrating [2] into our baseline analysis.
>
> Reference:
>
> [1] Torsional Diffusion for Molecular Conformer Generation
>
> [2] Do deep learning methods really perform better in molecular conformation generation?
>
> [3] Towards equilibrium molecular conformation generation with GFlowNets

---

### Official Review · Reviewer_3sE9 · 2023-11-01

**Soundness:** 2 fair
**Presentation:** 2 fair
**Contribution:** 2 fair
**Rating:** 3
**Confidence:** 5

**Summary:**

The paper proposes to model ligand torsions with LSTM.

Specifically, it 1) finds a way of extracting torsions as a sequence; 2) build torsion features with inputs and GNN features; and 3) tries to add diffusional training to the pipeline.

Experiments shows that the generated conformations are in good quality, in different senses.

**Strengths:**

1. Shown metrics are elevated compared with listed baselines.

2. The method is fast.

3. Not many research explored to model torsions as sequences so at least the paper is a novel exploration.

**Weaknesses:**

0. Overall the method does not seem to be interesting to the majority of the community. Pure Mol. Conf. Gen does not lead to any direct applications, and the general interest of the community is now moving to more challenging tasks. I would suggest the authors to demonstrate the usage of their model in some scenarios with real applications, such as trying their methods in docking.

1. Modelling torsion angles with LSTM is generally not a good idea. The permutation issues should be very carefully addressed, also the interactions in-between may not be fully explored (compared with using Transformer).

2. Recent work [1] shows that when used appropriately RDKit itself is a strongest baseline in conformation generation. It has better reported COV and MAT compared with this one. This hinders the significance of speeding up in this paper.

3. Technical details are not good. Sources of Figure data are not articulated (Fig2/4). Benmark performance, especially for GEOM-DRUGS, are not directly comparable to a majority of works in the community.

[1] G Zhou et al: Do Deep Learning Methods Really Perform Better in Molecular Conformation Generation?https://arxiv.org/pdf/2302.07061.pdf

**Questions:**

1. Why is LSTM a better choice than Transfomers? And how is the permutation issue dealed in this work? The canonical order derived from SMILES seems to be artificial. At least data augmentation tricks shall be leveraged.

2. Please explain the sources of data in Fig2/4.

3. I would expect some visualization to justify why the work is better than other baselines such as [1]

4. Analysis in steric clashes is required.

---

> ### Author Response · Authors · 2023-11-20
>
> Thanks for your valuable review, comments and questions:
>
> ## Weakness 0: the Significance and interest in MCG Task within our community:
> Here are some recent publications for 3D molecular conformer generation in top conferences: CGCF(ICML 2021), ConfVAE(ICML 2021), ConfGF(ICML 2021), GeoMol(NeurIPS 2021), GeoDiff(ICLR 2022), and Torsional Diffusion (NeurIPS 2022). From these publication records, we would argue that this task is popular in the ML community. At the same time, MCG is the fundamental in chemistry and biological field; it plays a role as the first steps in real-world research as mentioned in section 3.4 and Fig.2 of [1].
>
> ## Weakness 1 and Question 1 Sequential Model Choice:
> Thanks for pointing out this. We agree that transformer may better model the correlations than LSTM. The reason why we choose to use LSTM is its efficiency and effectiveness.
> LSTM is designed to effectively determine what information is retained or discarded at each step of a sequence through gate mechanism.
>
> Since efficiency is an important factor to consider in the molecular conformer generation task as discussed in the introduction section, LSTM can provide a quick solution in this setting. However, it's a promising future work to try out the transformer model in our model.

---

> ### Author Response · Authors · 2023-11-20
>
> ## Weakness 2 and 3 (performance part), Performance:
> (a) Different thresholds: The COV threshold are different between our paper and [3]. We follow [2], and [2] is the most recent SOTA method used a harder metric and got the highest performance in all metrics among all previous methods. Yet [3] and some previous works use $\delta = 1.25 \AA$. If [3] use 0.75 as threshold, the comparison results are as below:
>
>
> The COV threshould in this table is $\delta = 0.75\AA$, not $1.25\AA$ like most previous work. Note: Chemical Properties use $max (32, 2N_{ref})$ conformers, randomly selected from generated conformers; therefore for TorSeq, values we report in these chemical property metrics are the same.
>
> | Method                 | Prec COV $\uparrow$ | Prec AMR $\downarrow$ | $E$  | $\mu$ | $\Delta \epsilon$ | $E_{min}$ | Recall COV $\uparrow$ | Recall AMR $\downarrow$ | N_{ref} | Average runtime |  Total runtime  |
> |------------------------|:-------------------:|:---------------------:|:----:|:-----:|:-----------------:|:---------:|:---------------------:|:-----------------------:|:-------:|:-------:|:-------:|
> | Rdkit+ETKDG(Chem)      |       33.3 \%       |          0.89         | 0.81 |  0.52 |        0.75       |    1.16   |         28.6\%        |           1.00          |    2    |   0.05  |   0.10  |
> | Omega(Chem)            |       30.8 \%       |          0.85         | 0.68 |  0.66 |        0.68       |    0.69   |         54.6\%        |           0.76          |    2    |   0.10  |   0.28  |
> | CVGAE(Natural 2019)    |                     |          2.41         |      |       |                   |           |                       |           2.99          |    2    |         |         |
> | GraphDG(ICML 2020)     |                     |          2.42         |      |       |                   |           |                       |           1.93          |    2    |         |         |
> | CGCF(ICML 2021)        |                     |          1.82         |      |       |                   |           |                       |           1.22          |    2    |         |         |
> | ConfVAE(ICML 2021)     |                     |          1.82         |      |       |                   |           |                       |           1.14          |    2    |         |         |
> | ConfGF(ICML 2021)      |                     |          1.69         |      |       |                   |           |                       |           1.16          |    2    |         |         |
> | DGSM(NeurIPS 2021)     |                     |                       |      |       |                   |           |                       |           1.00          |    2    |         |         |
> | GeoMol(NeurIPS 2021)   |        36.4\%       |          0.84         | 0.42 |  0.34 |        0.59       |    0.40   |         41.4\%        |           0.83          |    2    |   0.19  |   0.38  |
> | GeoDiff(ICLR 2022)     |        14.5\%       |          1.09         | 0.31 |  0.35 |        0.89       |    0.39   |         37.8\%        |           0.81          |    2    |  505.97 | 1011.94 |
> | DMCG(TMLR 2022)        |                     |          0.88         |      |       |                   |           |                       |           0.79          |    2    |         |         |
> | Rdkit + Clustering [3] |        26.5\%       |          1.01         |      |       |                   |           |                       |           0.78          |    30   |   0.05  |   1.50  |
> | Tor.Diff(NeurIPS 2022) |        56.9\%       |          0.73         | 0.22 |  0.35 |        0.54       |    0.13   |         80.0\%        |           0.57          |    2    |   7.63  |  15.32  |
> | TorSeq (2N)            |        53.8\%       |          0.74         | 0.24 |  0.29 |        0.40       |    0.13   |         56.2\%        |           0.75          |    2    |   0.13  |   0.26  |
> | TorSeq (10N)           |        54.0\%       |          0.74         | 0.24 |  0.29 |        0.40       |    0.13   |         73.9\%        |           0.61          |    10   |   0.13  |   1.30  |
> | TorSeq (50N)           |        54.4\%       |          0.74         | 0.24 |  0.29 |        0.40       |    0.13   |         84.1\%        |           0.53          |    50   |   0.13  |   6.50  |
>
> (b) In fact [3] generate $30N_{ref}$ conformers, and use clustering to remove duplication, which leads to higher performance in the recall metric.
> We have mentioned and discuss this in section 4.3 and Figure 5 of our paper.
> We report a more detailed results of our approach in Table above, in recall metric, we can get better performance than previous methods within fewer computational cost.
>
> On page 4 of the [3] section 3. "Specifically, our sampling process involves the uniform, geometric, and energy samplers, which are used in a ratio of 1:1:4,
> respectively. The number of energy samples, denoted by $N_e$, is determined by the formula $N_e = min (20N_{ref}, 2000)$".

---

> ### Author Response · Authors · 2023-11-20
>
> ## Weakness 3 (figure detail part) and Question 2, Figure Detail:
> We describe Fig.2 in section 3.1, the Fig.2 left is the circular-circular correlation values among torsion angles in all conformers of all molecules in training set; the Fig.2 right is three random selected molecule to show the correlation relationship heatmap, SMILES of these molecule is list on the top side. Fig.4 is the distribution a torsion angle of conformers in a random selected molecule from training set. We use this to describe the angle near $\pi$ may suffer vanishing gradient. This is a common torsion angle distribution in this task, because a torsion angle locate near pi meet the requirement of the theoretical minimum energy. The most famous and basic example is the degree of torsion angle cccc in anti-staggered butane.
>
> ## Question 3 and 4 Visualization and Steric Clashes Analysis:
>
> 1. We updated the visualization in the Appendix. Use the molecules selected by [4] for comparison as fairly as possible
>
> 2. We update the Analysis of steric clashes in the Appendix. Compared with torsional diffusion, the SOTA method.
>
>
> Reference:
>
> [1] Generative models for molecular discovery: Recent advances and challenges
>
> [2] Torsional Diffusion for Molecular Conformer Generation
>
> [3] Do Deep Learning Methods Really Perform Better in Molecular Conformation Generation?
>
> [4] GeoMol: Torsional Geometric Generation of Molecular 3D Conformer Ensembles

---

> ### Comment · Reviewer_3sE9 · 2023-11-22
> **reply to rebuttal**
>
> Thank you for the reply, but there are still some concerns that are not resolved:
>
> 1) listing a bunch of publications in relative areas is not helpful in showing that MCG *is still* a major concern in the field of AI + molecule science. Indeed some insightful previous work in MCG shed light on new methods in more applicational scenarios such as molecule docking (protein-ligand prediction) and direct drug candidate generation, toying with GEOM-QM9 or GEOM-DRUGS to generate molecule conformations is no longer helpful, because the community already know how deep learning methods can help. A more persuasive defend in still studying MCG would be showing more directly applicational cases, or establishing more challenge benchmarks.
>
> 2) essentially, I don't quite know the advantage of TorSeq to Tor. Diff. As is stated by Reviewer Mhor, "The proposed method does not seem to outperform "Torsional Diffusion" as a standalone technique. While combining it with "Torsional Diffusion" yields better results than "Torsional Diffusion" alone, the increased runtime compromises its efficiency. Thus, the paper's claim of achieving both effectiveness and efficiency is not well-supported, as a trade-off exists between the two." If the statement of efficiency is the main idea of this paper then the accuracy should be more convincing; if the statement of the advantage of LSTM is the main idea then more comparisons should be done with studying backbones (eg comparing with Transformers under exact settings). The paper failed to do anything deeper than just playing with the datasets.
>
> 3) I don't know the specific meaning of benchmarking TorSeq (10/50N) to other baselines. Elevations in such setups are almost trivial. If the authors want to accurately benchmark precisions and recalls under 10/50N then baseline results should be extended to such scenarios as well.
>
> 4) questions wrt LSTM's permutation issues are not addressed. Also claiming LSTM is a faster implementation to Transformers also needs justification: since we are expecting <100 torsion angles then the parallel nature of transformer may even be more significant to its O(n^2) complexity. Empirically both LSTM and Transformers are fast enough to make efficiency not a blocking issue.
>
> 5) by Question 2 I was asking about the concrete protocol of deriving statistics in Fig2/4 to justify the rigour of a research paper, instead of inquiring about their meanings.
>
> Overall, the studied topic is not interesting, the method is not solid  / needs more justification, and the experiments are not impressive. I would not change the rating, and I find this is a consensus among all reviewers.

---

### Official Review · Reviewer_sEK1 · 2023-11-01

**Soundness:** 2 fair
**Presentation:** 2 fair
**Contribution:** 2 fair
**Rating:** 3
**Confidence:** 4

**Summary:**

this paper propose TorSeq for MCG task,which introduce an artificial torsional sequence enabling explicitly modeling interrelations among torsion angles. results in GEOM also show effectiveness and efficiency of the proposed methods.

**Strengths:**

propose a sequential based MCG method, which predict low energy torsion, also this method can intergate with torsion diffusion.

**Weaknesses:**

Autoregressive generation is less novelty, also experiment gain is not signicfiant enough ( compare with serval SOTA DL based models).

**Questions:**

As mentioned in recently works[https://arxiv.org/pdf/2310.14782.pdf], `a recent approach combining ETKDG with clustering (Zhou et al.,
2023), which was shown to outperform most existing machine learning methods in the low energy conformation generation task.`
I believe that a more in-depth discussion on this topic is warranted, particularly regarding the usefulness and informativeness of the datasets employed for general molecular conformation generation or try to generate pure GFN2-xTB level(semi-DFT as in GEOM-Drugs) conformation is enough?

---

> ### Author Response · Authors · 2023-11-20
>
> Thanks for your valuable review and comments!
>
> ## Weakness Novelty
> Our research introduces a novel approach centered on sequential modeling for torsion angles, which enables better torsion angle generation by considering previously generated torsion angles. This approach has shown to be effectiveness not only for autoregressive models but also for diffusion models empirically.
>
> Additionally, we present another innovation in addressing the issue of gradient vanishing, a common challenge in the direct reconstruction of torsion angles. Our solution offers a strategic method to analyze and mitigate this problem, enhancing the model's ability to learn and reconstruct torsion angles more accurately.

---

> ### Author Response · Authors · 2023-11-20
>
> ## Weakness Performance
> There's a misunderstanding on the performance comparison with previous methods. Unfair comparison exists between the paper [1], and SOTA methods, including our work.
>
> (a) Different thresholds: The COV threshold are different between our paper and [1]. We follow [2], and [2] is the most recent SOTA method used a harder metric and got the highest performance in all metrics among all previous methods. Yet [1] and some previous works use $\delta = 1.25 \AA$. If [1] use 0.75 as threshold, the comparison results are as below:
>
> The COV threshould in this table is $\delta = 0.75\AA$, not $1.25\AA$ like most previous work. Note: Chemical Properties use $max (32, 2N_{ref})$ conformers, randomly selected from generated conformers; therefore for TorSeq, values we report in these chemical property metrics are the same.
>
> | Method                 | Prec COV $\uparrow$ | Prec AMR $\downarrow$ | $E$  | $\mu$ | $\Delta \epsilon$ | $E_{min}$ | Recall COV $\uparrow$ | Recall AMR $\downarrow$ | N_{ref} | Average runtime |  Total runtime |
> |------------------------|:-------------------:|:---------------------:|:----:|:-----:|:-----------------:|:---------:|:---------------------:|:-----------------------:|:-------:|:-------:|:-------:|
> | Rdkit+ETKDG(Chem)      |       33.3 \%       |          0.89         | 0.81 |  0.52 |        0.75       |    1.16   |         28.6\%        |           1.00          |    2    |   0.05  |   0.10  |
> | Omega(Chem)            |       30.8 \%       |          0.85         | 0.68 |  0.66 |        0.68       |    0.69   |         54.6\%        |           0.76          |    2    |   0.10  |   0.28  |
> | CVGAE(Natural 2019)    |                     |          2.41         |      |       |                   |           |                       |           2.99          |    2    |         |         |
> | GraphDG(ICML 2020)     |                     |          2.42         |      |       |                   |           |                       |           1.93          |    2    |         |         |
> | CGCF(ICML 2021)        |                     |          1.82         |      |       |                   |           |                       |           1.22          |    2    |         |         |
> | ConfVAE(ICML 2021)     |                     |          1.82         |      |       |                   |           |                       |           1.14          |    2    |         |         |
> | ConfGF(ICML 2021)      |                     |          1.69         |      |       |                   |           |                       |           1.16          |    2    |         |         |
> | DGSM(NeurIPS 2021)     |                     |                       |      |       |                   |           |                       |           1.00          |    2    |         |         |
> | GeoMol(NeurIPS 2021)   |        36.4\%       |          0.84         | 0.42 |  0.34 |        0.59       |    0.40   |         41.4\%        |           0.83          |    2    |   0.19  |   0.38  |
> | GeoDiff(ICLR 2022)     |        14.5\%       |          1.09         | 0.31 |  0.35 |        0.89       |    0.39   |         37.8\%        |           0.81          |    2    |  505.97 | 1011.94 |
> | DMCG(TMLR 2022)        |                     |          0.88         |      |       |                   |           |                       |           0.79          |    2    |         |         |
> | Rdkit + Clustering [1] |        26.5\%       |          1.01         |      |       |                   |           |                       |           0.78          |    30   |   0.05  |   1.50  |
> | Tor.Diff(NeurIPS 2022) |        56.9\%       |          0.73         | 0.22 |  0.35 |        0.54       |    0.13   |         80.0\%        |           0.57          |    2    |   7.63  |  15.32  |
> | TorSeq (2N)            |        53.8\%       |          0.74         | 0.24 |  0.29 |        0.40       |    0.13   |         56.2\%        |           0.75          |    2    |   0.13  |   0.26  |
> | TorSeq (10N)           |        54.0\%       |          0.74         | 0.24 |  0.29 |        0.40       |    0.13   |         73.9\%        |           0.61          |    10   |   0.13  |   1.30  |
> | TorSeq (50N)           |        54.4\%       |          0.74         | 0.24 |  0.29 |        0.40       |    0.13   |         84.1\%        |           0.53          |    50   |   0.13  |   6.50  |

---

> ### Author Response · Authors · 2023-11-20
>
> (b) In fact [1] generate $30N_{ref}$ conformers, and use clustering to remove duplication, which leads to higher performance in the recall metric.
> We have mentioned and discuss this in section 4.3 and Figure 5 of our paper.
> We report a more detailed results of our approach in Table above, in recall metric, we can get better performance than previous methods within fewer computational cost.
>
> On page 4 of the [1] section 3. "Specifically, our sampling process involves the uniform, geometric, and energy samplers, which are used in a ratio of 1:1:4, respectively. The number of energy samples, denoted by $N_e$, is determined by the formula $N_e = min (20N_{ref}, 2000)$"
>
>
> ## Question 1
> Thanks for your question.
>
> The GEOM-DRUGS dataset has about 300k molecules with about 100 conformers per molecule, i.e., about 30M conformers.
> The GEOM-DRUGS use the SOTA molecular dynamics method, CREST (Pracht et al., 2020a) to generate conformers. Although small errors may be inevitable, this is the highest accuracy molecule conformers dataset humans can get now. The fatal disadvantage of CREST is the huge computational cost, which is unsuitable for high-throughput applications. The computational cost problem of molecule dynamics is the root motivation of the molecule conformation generation task.
>
> Reference:
>
> [1] Do Deep Learning Methods Really Perform Better in Molecular Conformation Generation?
>
> [2] Torsional Diffusion for Molecular Conformer Generation

---

### Official Review · Reviewer_Xdkx · 2023-11-01

**Soundness:** 2 fair
**Presentation:** 2 fair
**Contribution:** 2 fair
**Rating:** 3
**Confidence:** 4

**Summary:**

This paper showed that torsion angles in a molecular is related and proposed a model named TorSeq which uses LSTM for sequential torsion angle prediction. Compared with the torsional diffusion's backbone, authors argued that LSTM models can explicitly model the interrelations between the torsion angles by imitation of decomposition of conditional probabilities. Such highlight of the model of interrelations can enhance the performance. Experiments show that TorSeq outperforms multiple baseline methods in terms of both effeciency and accuracy.

**Strengths:**

1. Efficacy Demonstrated through Experimental Results: Empirical evidence substantiates TorSeq's exceptional efficiency when juxtaposed with other extant methods. This compelling empirical validation underscores the robustness of TorSeq as a pertinent solution within the domain of interest.

2. Innovative Resolution of the Gradient Vanishing Problem: TorSeq introduces an interesting strategy for mitigating the gradient vanishing problem, representing a novel and noteworthy contribution to the field. The simplicity of this method belies its effectiveness in surmounting a challenge that has previously posed significant impediments to progress.

**Weaknesses:**

The primary contribution of this research paper lies in the proposition of an enhanced method for the more accurate prediction of torsion angles, which takes into account their interrelations. A mere reliance on the decomposition of conditional probability and the simplistic application of LSTM appears insufficient in addressing the intricacies inherent in modeling torsion angles. Consequently, there exist notable limitations and inadequacies associated with such an approach, which warrant discussion and exploration.

1. About explicit and implicit: The authors contend that earlier models have predominantly employed implicit mechanisms to capture the correlations among torsion angles. In contrast, the utilization of a RNN to emulate conditional probability enables the explicit modeling of these correlations. It is apparent that the principal disparity between the proposed model and its predecessors lies in the emphasis on explicit modeling through conditional probability. Nonetheless, it is noteworthy that previous models can also be interpreted as involving conditional prediction and, consequently, the explicit modeling of such correlations. A more comprehensive analysis is warranted to delineate the differentiating aspects of the proposed model in comparison to the antecedent approaches.

2. About highlights on adjacent torsion angles: The authors have emphasized the importance of highlighting the interdependence among torsion angles, particularly those that are adjacent to one another. However, a thorough examination of the proposed model reveals a lack of explicit emphasis on this characteristic. As an illustrative example, one can refer to Figure 3, where the authors suggest that the relationship between the torsion angles $\tau_4$ and $\tau_7$ should be accentuated, given their adjacency to the chemical moiety c1nnc. Nevertheless, in the case of LSTM employed in the model, there is a paucity of specific information to robustly establish a strong connection between $\tau_4$ and $\tau_7$. Given that most adjacent torsion angles in a molecule are also adjacent in the torsion angle sequence, LSTM may inadvertently overlook the correlation between $\tau_4$ and $\tau_7$ in such instances. This leads to the conclusion that the explicit representation of torsion angle correlations in the proposed model is arguably inadequate. It is posited that the integration of an attention mechanism may address these concerns, yet the paper lacks a detailed analysis of this potential solution.

3. About conditional probability: While Equation 2 is unquestionably mathematically sound, it fails to account for the potential presence of a ``dominant'' torsion angle within a molecule. It is conceivable that certain torsion angles exert a significant influence on the overall structure, while others exhibit limited dependencies to each other. In scenarios where such a dominant torsion angle is positioned at the rear end of the torsion angle sequence, the predictive accuracy for other torsion angles may be compromised.

**Questions:**

1. Object of modeling: To create conformations, it is essential to have access to both torsion angles and local structures. Equation 1 posits the primary objective as the generation of torsion angles based on a given molecular graph. This raises the question of how the local structures are acquired prior to predicting the torsion angles. Is it necessary to employ a tool like RDKit to generate these local structures, akin to the methodology employed in Torsional Diffusion?

2. Distinguishing prediction and generation tasks: In Equation 1, the author characterizes the task as a generative one. However, ensuring that the generated torsion angles faithfully adhere to the distribution of the dataset presents a significant challenge. How can we establish the veracity of the claim that the generated torsion angles indeed conform to the dataset's distribution?

---

> ### Author Response · Authors · 2023-11-20
>
> Thanks for your valuable review, comments and questions.
>
> ## Weakness One - More analysis about the necessity of TorSeq:
> In the task of Molecular Conformation Generation (MCG), sequential modeling is a crucial complement. Molecular modeling is usually limited to 3 to 4 GNN layers, which is insufficient for comprehending multiple-hop topological relationships.
> We also update this discussion into Appendix Discussion Part. We think this can more comprehensive describe of the necessity of sequence modeling for MCG tasks.
>
> ## Weakness Two - About highlights on adjacent torsion angles:
> LSTM is designed to effectively manage long-term information. LSTM achieve this through their unique gating mechanisms, which judiciously determine what information is retained or discarded at each step of a sequence. The primary focus of this paper is to explore the importance of sequentially modeling torsion angles. However, we deeply agree that investigating more sophisticated model designs that enhance the sequential modeling of torsion angles presents an exciting avenue for future research.
>
> ## Weakness Three - About conditional probability
> The torsion angle representation is acquired by combining the graph feature, which includes the topological feature among atom nodes or motif-level node feature. Using these features, we believe the LSTM gates can learn the "dominant" torsion angle.
> Also, bidirectional LSTM can ensure that all torsion angles will fully influence the sequence. For example, $\tau_7$ in Fig.3 is not expected to influence the sequence in forward LSTM, but it will be the first token to be considered in backward LSTM.
>
> In SMILES notation, atoms are represented alongside numbers that indicate broken ring structures, which transform the molecular graph into a tree. Brackets are used to denote branches. Therefore, in the implementation of our proposed method, the canonical SMILES is akin to a Depth-First Search (DFS) algorithm, which is consistent with the atomic number in both of used datasets.
>
> In the early stage of our research, we experimented with two different DFS algorithms. The first, which we refer to as DFS-GEOM, follows the dataset’s default atomic number. The second approach prioritizes shorter branches in the molecular tree. In cases where branch lengths are the same, we prioritize atoms with smaller atomic number. These two DFS approaches yield different sequencing patterns for the same molecule, though the overall experimental results were similar. This observation will be revised in the final version of our paper. Here, we retrain and test the DFS-GEOM with final version model. Results are as detailed in the table below. Using different DFS algorithms does not significantly change the performance.
>
> |                     |  |    RMSD AMR $\downarrow$   |           |        |
> |---------------------|:--------:|:------:|:---------:|:------:|
> |                     |  Recall     |    | Precision           |
> | DFS Method          |   Mean   | Median |    Mean   | Median |
> | GEOM-DFS            |  0.768 |  0.745 |   0.814   |  0.758 |
> | Short branch first  |   0.773  |  0.748 |   0.806   |  0.744 |
>
> ## Question One - Object of modeling:
>
> Thanks for your question!
> Following torsional diffusion, we use the local generated by RDKit as well. We mentioned this On page 5, section 3.4, ”Our approach employs changing torsion angles by following the same approach as (Jinget al., 2022), which consequently preserves the SE(3) invariance property.”
> We will add more details to further clarify this part in the final version.

---

> ### Author Response · Authors · 2023-11-20
>
> ## Question Two - Distinguishing prediction and generation task:
> 1. Theoretically, during our training process, the loss of torsion angle is not calculated independently. Taking Figure 2 as an example, we will use the total loss of $\tau_1, \tau_ 2,... \tau_8$ of each set of reference conformer and predicted conformer so that we can ensure that each set of torsion angles faithfully trained on the torsion angle of the actual conformer. Also, EMD will ensure there is one-to-one correspondence for each pair of references and predicted conformers during training.
>
>
> |              |  |    Precision    |       |        |
> |--------------|:---------:|:------:|:-----:|:------:|
> |              |  Coverage |        |  AMR  |        |
> | DFS Method   |    Mean   | Median |  Mean | Median |
> | random $\tau$ |    18.2   |   4.0  | 1.228 |  1.217 |
> | TorSeq       |    52.6   |  53.8  | 0.806 |  0.744 |
>
> 2. We also empiracally verify the alignment of distributions. We use the RMSD precision COV and AMR metrics to evaluate the generation results, which are commonly used evaluation metrics in the community. These metrics have been validated in previous work [1]. If the torsional angle does not fit the distribution of the dataset, in one of these cases, random torsion angle, the conformer show high RMSD.
> Table 8 on page 27 of [1]. It demonstrates poor geometry precision performance when the torsion angle is randomly set, i.e., the torsion angle does not match the ground truth distribution.
>
> Reference:
> [1] Torsional Diffusion for Molecular Conformer Generation

---

### Meta-Review · Area_Chair_oybC · 2023-12-06

**Metareview:**

The paper proposes a sequential LSTM-based generative model, TorSeq, of torsional angles for small and mid-sized molecules. The main strengths pointed out by the reviews are that TorSeq has a very low runtime and compares favourably in performance to non-diffusion-based methods. When TorSeq is combined with Torsional Diffusion (Jing et al., 2022), it also outperforms diffusion-based methods on the GEOM-DRUGS and GEOM-QM9 datasets. Also, the paper proposes a way of addressing address vanishing gradients when expressing torsional angles through the transformation $\pi \tanh(\cdot)$,

Several reviewers raised issues regarding modelling torsional angles sequentially using an LSTM, and while one reviewer appreciated the novelty of this approach, others raised concerns about its limited novelty and relevance. Also, several reviewers found it a weakness that the proposed model only performed well when combined with an existing diffusion-based mode.

**Justification For Why Not Higher Score:**

There is a strong consensus in concerns among the reviewers and they all recommend the paper to be rejected.

**Justification For Why Not Lower Score:**

N/A

---

### Decision · Program_Chairs · 2024-01-16

Reject